# Breast cancer-related lymphedema and recurrence of breast cancer: Protocol for a prospective cohort study in China

**Linli Zhuang**[1]*, **Qian Chen**[2], **Huaying Chen**[3], **Xuemei Zheng**[3], **Xia Liu**[4], **Zhenzhen Feng**[3,5], **Shaoyong Wu**[3], **Li Liu**[3], **Xiaolin Shen**[1]

**1** Department of Rheumatology and Immunology, West China Hospital of Sichuan University, Chengdu, Sichuan, China, **2** Center of Biostatistics, Design, Measurement and Evaluation, Department of Clinical Research Management, West China Hospital of Sichuan University, Chengdu, Sichuan, China, **3** Department of Head and Neck Oncology, Cancer Center, West China Hospital, Sichuan University, Chengdu, China, **4** Department of Abdominal Oncology, Cancer Center, West China Hospital, Sichuan University, Chengdu, China, **5** Unit of Cancer Day Care, West China Hospital, Sichuan University, Chengdu, China

* 1637194326@qq.com

## Abstract

### Introduction

The primary aim is to determine the factors associated with breast cancer-related lymphedema and to identify new associated factors for the recurrence of breast cancer and depression. The secondary objective is to investigate the incidence of breast cancer-related events (breast cancer-related lymphedema, recurrence of breast cancer, and depression). Finally, we want to explore and validate the complex relationship among multiple factors influencing breast cancer complications and breast cancer recurrence.

### Patients and methods

A cohort study of females with unilateral breast cancer will be conducted in West China Hospital between February 2023 and February 2026. Breast cancer survivors in the age range of 17–55 will be recruited before breast cancer surgery. We will recruit 1557 preoperative patients with a first invasive breast cancer diagnosis. Consenting breast cancer survivors will complete demographic information, clinicopathological factors, surgery information, baseline information, and a baseline depression questionnaire. Data will be collected at four stages: the perioperative stage, chemotherapy therapy stage, radiation therapy stage, and follow-up stage. Data including the incidence and correlation of breast cancer-related lymphedema, breast cancer recurrence, depression, and medical cost will be collected and computed through the four stages above. For every statistical analysis, the participants will be classified into two groups based on whether they develop secondary lymphedema. Incidence rates of breast cancer recurrence and depression will be calculated separately for groups. Multivariate logistic regression will be used to determine whether secondary lymphedema and other parameters can predict breast cancer recurrence.

**Data Availability Statement:** Deidentified research data will be made publicly available when the study is completed and published.

**Funding:** This work was supported by the Sichuan Science and Technology Support Program (2015SZ017). The funders had and will not have a role in study design, data collection and analysis, decision to publish, or preparation of the manuscript.

**Competing interests:** The authors have declared that no competing interests exist.

## Discussion

Our prospective cohort study will contribute to establishing an early detection program for breast cancer-related lymphedema and recurrence of breast cancer, which are both associated with poor quality of life and reduced life expectancy. Our study can also provide new insights into the physical, economic, treatment-related and mental burdens of breast cancer survivors.

## Introduction

Breast cancer-related lymphedema (BCRL) is closely related to a high risk of cancer-related treatment, breast cancer recurrence, and disability [1]. In breast cancer patients, secondary lymphedema is a symptom cluster encompassing progressive upper-limb swelling, skin changes, limb numbness, pain and discomfort, restricted range of motion, and nonpitting edema [2]. These would decrease physical and mental function, increase disease burden and increase financial costs. It is a common complication in breast cancer patients [3–6]. Considering the establishment of a baseline of the upper limb and tracking changes, a screening cohort of secondary lymphedema may be the optimal solution to develop screening programs and evaluate the prevalence of breast cancer-related lymphedema in China.

Breast cancer recurrence (BCR) is a negative outcome in patients with breast cancer after either mastectomy or breast-conserving therapy (BCT) [7]. Breast cancer can recur locally, regionally, and/or at distant metastatic sites [8]. It is a hammer blow for breast cancer survivors [9]. It directly threatens breast cancer survivors' life expectancy and decreases their quality of life. BCRL and BCR share risk factors such as higher BMI, pathological type, breast cancer location, tumor node metastasis(TNM) staging and comorbidities, and treatment-related factors (surgery, radiotherapy, and chemotherapy) [10]. Early detection helps implement personal and precise medical precautions for patients with secondary lymphedema (or breast cancer recurrence).

Up to 30% of breast cancer survivors have depressive symptoms [11]. Depressive affect includes emotional lability, irritability, and social withdrawal [12]. Depression is an emotional difficulty associated with an increased disease burden and treatment-related cost and a decreased ability to cope with breast cancer. Fear of cancer recurrence also directly affects mental health, especially anxiety and depression [13].

These three factors are significant risk factors for the disease burden of breast cancer. The disease burden of breast cancer is an area of concern [14]. The disease burden of breast cancer is a multidimensional construct comprising surgery, oncology-related treatment, drug costs and depression, and other elements destroying everyday life [15]. Therefore, a comprehensive screening plan that is feasible and responsive to reality may alleviate the symptom burden and reduce the economic burden of breast cancer patients.

Based on the retrospective data of the electronic medical record system in West China Hospital, we found a 3-year BCRL incidence rate of 17.57% after breast surgery or breast reconstruction in patients with breast cancer. The rate of BCR occurrence was 15.53% at three years. A previous study found that breast cancer recurrence may harm breast cancer-related lymphedema. However, whether BCRL is associated with the risk of breast cancer recurrence is uncertain. Moreover, there is a lack of evidence regarding the correlation between BCR, BCRL, and depressive symptoms, and there is also inadequate knowledge regarding the disease burden of breast cancer. This study aims to establish and evaluate whether a comprehensive

screening plan for breast cancer may catch cancer early and improve the health situation and prognosis.

## Patients and methods

This study is a prospective cohort study of breast cancer survivors. This study aimed to investigate and evaluate the incidence and correlation of BCRL, BCR, and depression. The primary outcome measures include the incidence and correlation of BCRL and BCR within three years. Choice of the time frame because three years is a high-incidence period of BCRL and BCR. The secondary outcome is depression.

### Ethics approval and consent to participate

Our study period is between February 2023 and February 2026. The study was approved by the Ethics Committee of West China Hospital (Approval No. of ethics committee: 2019 Annual Review No. 664). Registration number: ChiCTR1900025534 (Name of the registry: Chinese Clinical Trial Registry (ChiCTR) http://www.chictr.org.cn/)

Our cohort study is in line with the international ethical principles and ethical processes of West China Hospital. All participants provided written informed consent. Breast cancer survivors will be informed that their identifying information will be confidential before participating in our cohort study. Fully informed consent will be obtained because the total duration of the study will span almost three years. Participants can withdraw at any time from the study. Fig 1 shows the participant flow.

### Study sample

We will conduct a prospective cohort study of females with unilateral breast cancer in West China Hospital. We will recruit breast cancer patients between the ages of 18 and 55 years who will receive unilateral radical mastectomy or modified radical mastectomy, postoperative chemotherapy, and radiotherapy. Patients with breast cancer will be the first breast cancer diagnosed in West China Hospital, and the data on surgery and treatment will be extracted from the electronic medical record system. This cohort study will commence one week before breast cancer surgery and conclude after the three-year follow-up period.

Our study will need 1557 preoperative patients with a first-time invasive breast cancer diagnosis. Patients must meet the following criteria for inclusion.

- Female subjects with ages older than 18 years and less than 55 years.

- Understand and sign the informed consent.

- Recruited in West China Hospital.

- First diagnosis of invasive breast cancer.

- Breast cancer patients will receive either radical or modified radical mastectomy.

  There are four exclusion criteria; they are described briefly below.

- Patients with bilateral breast cancer

- Patients had previously received neoadjuvant radiotherapy treatment before.

- Patients could not complete all the treatments and 3-year follow-up.

- There is significant asymmetry in the size of the upper extremities.

- The patients had cardiac dysfunction or renal and hepatic insufficiency.

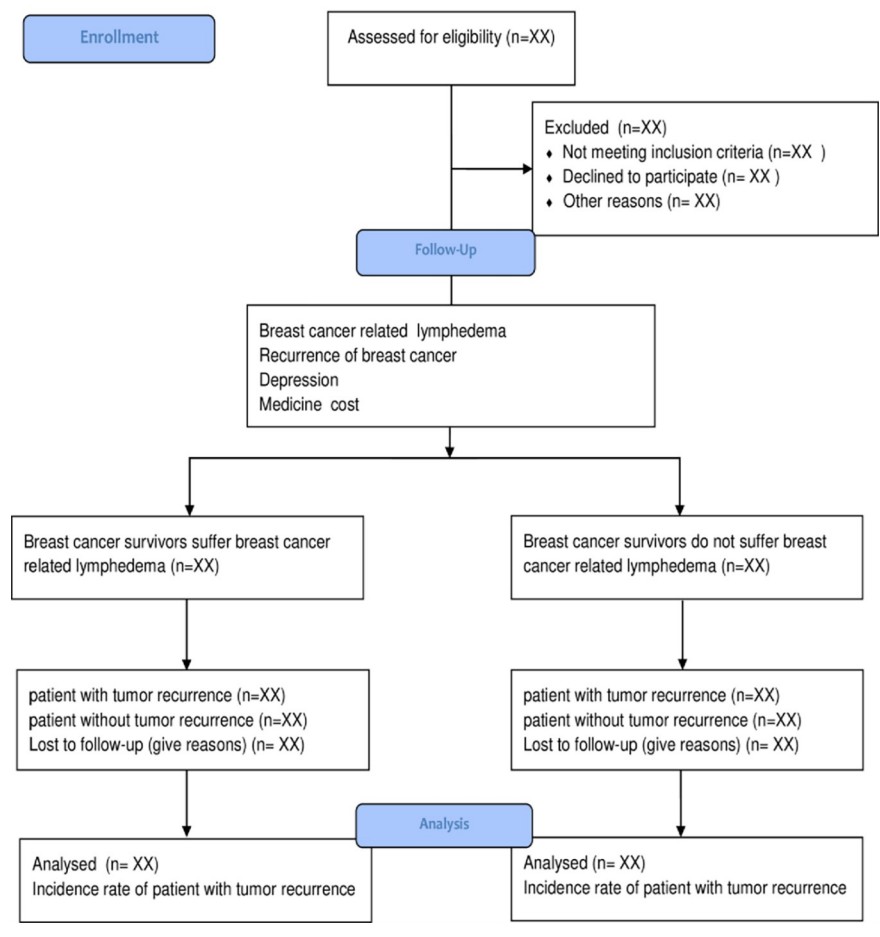

**Fig 1. CONSORT 2010 flow diagram.**

**Study procedure.** *Baseline assessment.* Basic demographic and treatment information will be collected through an electronic medical record system. Our study will gather self-reported demographic information, including age, sex, marital status, education level, etc. We will obtain surgery and treatment information, including surgery date, surgery type (axillary lymph node dissection or sentinel lymph node dissection), the total number of dissected lymph nodes, chemotherapy regimen, radiotherapy regimen, and cycle number of treatment (chemotherapy/radiotherapy).

*Ongoing assessment.* Table 1 shows the standard assessment process of the follow-up assessment of our study. Clinical evaluations will include evaluations of BCRL, RBC, and depression. At each follow-up visit, patients will receive an upper-limb volume assessment and physical

**Table 1. The standard assessment process in cohort study—time points and primary outcomes.**

| Time point | BMI | Patients with secondary lymphedema or not | Patients with oncology recurrence or not | Beck Depression Inventory II | Medicine cost |
|---|---|---|---|---|---|
| **Perioperative stage** | | | | | |
| before surgery | √ | / | / | √ | √ |
| **Chemotherapy therapy stage** | | | | | |
| circle 1 | √ | √ | √ | √ | √ |
| circle 2 | √ | √ | √ | √ | √ |
| circle 3 | √ | √ | √ | √ | √ |
| circle(4.5.6 . . .n- 1) | √ | √ | √ | √ | √ |
| circle(n)—last circle | √ | √ | √ | √ | √ |
| **Radiation therapy stage** | | | | | |
| before radiation therapy | √ | √ | √ | √ | √ |
| after radiation therapy | √ | √ | √ | √ | √ |
| **Follow-up stage** | | | | | |
| 9th month(after surgery) | √ | √ | √ | √ | √ |
| 1st year(after surgery) | √ | √ | √ | √ | √ |
| 15th month(after surgery) | √ | √ | √ | √ | √ |
| 18th month(after surgery) | √ | √ | √ | √ | √ |
| 21th month(after surgery) | √ | √ | √ | √ | √ |
| 2nd years(after surgery) | √ | √ | √ | √ | √ |
| 27th month(after surgery) | √ | √ | √ | √ | √ |
| 30th month(after surgery) | √ | √ | √ | √ | √ |
| 33th month(after surgery) | √ | √ | √ | √ | √ |
| 3rd year(after surgery) | √ | √ | √ | √ | √ |

and imaging examination [16]. The research nurse will help them complete the Beck Depression Inventory-II questionnaire about patients' mental status.

**Diagnosis standards.** Diagnosis of BCRL: We used a 200 ml limb volume difference as the diagnostic standard. The 200 ml limb volume difference standard is a universal criterion recommended by the 2020 Consensus Document of the International Society of Lymphology [2]. The limb volume difference will be measured with a Perometer device(an automated optoelectronic measurement of arm circumferences) [17].

Diagnosis of RBC: Physical examination and magnetic resonance imaging (MRI) for screening surveillance of breast cancer survivors. Breast cancer survivors who had biopsy-proven recurrence of breast cancer at presentation documented by fine-needle aspiration biopsy or core needle biopsy [18–21].

Diagnosis of depression: The screening tool will be the Beck Depression Inventory-II. We used the Beck Depression Inventory-II rubric; The scores are subdivided into the following grades: minimal(0–13 points), mild(14–19 points), moderate(20–28 points), and severe (greater than 29 points) [22, 23].

**Record.** We will record the outcomes in detail, including the onset time, diagnosis time, symptoms, adherence, and medicine cost. The incidence in each stage will be calculated and used to explore complex relationships among the outcomes.

## Data collection

Data will be collected at four stages: the perioperative stage, chemotherapy therapy stage, radiation therapy stage, and follow-up stage. Having completed the previous phase, we move on to

the next. The total expenditure on healthcare will be computed for each treatment period. All authors could access information that could identify individual participants during data collection and statistical analysis.

**First stage—perioperative stage.** Data will be collected at the time of the perioperative stage, including demographic data (sex, age, race, ethnicity, marital status, socioeconomic status, educational status, and BMI), clinicopathological factors (pathological type, breast cancer location, TNM staging, and comorbidities), surgery information (date of diagnosis, date of surgery, type of surgery, type of surgical incision, lymph node management, the total number of dissected lymph nodes), and baseline information (upper-limb volume difference, scale results of Beck Depression Inventory-II). Patient information in the perioperative stage will be collected through the electronic medical record after obtaining written informed consent. We will manage to build a follow-up database preoperatively. We will use this follow-up database for baseline and data collection, transfer, and storage.

**Secondary stage—chemotherapy therapy stage.** Data will be collected at the time of the chemotherapy therapy stage, including the number of chemotherapy cycles, body mass index (BMI), upper limb volume measurement, assessment of tumor recurrence, and scale results of the Beck Depression Inventory-II. Once they receive the first round of chemotherapy, participants will be followed up in a regular oncology ward each cycle. Data collectors will conduct breast cancer-related lymphedema surveillance through upper-limb volume measurements. For patients who increase the upper-limb volume difference by more than 200 ml, secondary lymphedema is confirmed and diagnosed by a physician and physical therapist. Data will be collected at the time of the radiation therapy stage including the total number of radiation therapy cycles, body mass index (BMI), upper limb volume measurement, assessment of tumor recurrence, and scale results of the Beck Depression Inventory-II.

**Third stage—radiation therapy stage.** Radiation therapy cycles terminate at relatively short times. Therefore, data on the main factors will only be collected before radiation therapy cycles and after stopping the planned radiation treatment.

**Last stage—follow-up stage.** The follow-up stage is also the last stage of our cohort study. After finishing radiation and chemotherapy treatment, patients will be followed until 3 years after breast cancer surgery. Postoperative breast cancer patients are exposed to a chance of recurrence. The implementation of guideline-recommended follow-up plans for breast cancer patients is necessary. The patients in our study will be seen for follow-up visits once every three months [24]. Data will be collected at the time of the follow-up stage including BMI, upper limb volume measurement, assessment of tumor recurrence, and scale results of the Beck Depression Inventory II. Detailed cohort protocols in the four research stages are presented in Table 2.

## Quality assurance

To ensure the uniformity of data, all the nurses in our study will undergo unified training about the measurement of arm volume and how to provide guidance when patients complete the questionnaire. Arm volume will be measured twice, in the morning and at night, and the average of two measurements will be used in the analyses to minimize errors. All oncologists will receive unified training, including magnetic resonance imaging (MRI) analysis and diagnosis of breast cancer recurrence. Moreover, loss to follow-up will continue to hinder the data completeness of our cohort study. Therefore, the computer and application are programmed to alert patients by telephone and message before one day when breast cancer patients need to complete the follow-up.

**Table 2. Detailed cohort protocols in four research stages.**

| Perioperative stage | |
|---|---|
| Demographic data | gender, age, race, ethnicity, marital status, socioeconomic status, educational status and body mass index (BMI) |
| Clinicopathological factors | pathological type, breast cancer location, TNM staging and comorbidities |
| Surgery information | date of diagnosis, date of surgery, type of surgery, type of surgical incision, Lymph node management, the total number of dissected lymph nodes |
| Baseline information | arm circumferences, scale results of Beck Depression Inventory-II |
| Financial information | treatment related cost |
| **Chemotherapy therapy stage** | |
| Time points information | the number of chemotherapy cycles |
| Main information | body mass index(BMI), arm volume measurement (secondary lymphedema), assessment of tumor recurrence, scale results of Beck Depression Inventory-II |
| Financial information | treatment related cost |
| **Radiation therapy stage** | |
| Time points information | total number of radiation therapy cycles |
| Main information | body mass index (BMI), arm volume measurement (secondary lymphedema), assessment of tumor recurrence, scale results of Beck Depression Inventory-II |
| Financial information | treatment related cost |
| **Follow-up stage** | |
| Time points information | the number of follow-up cycles |
| Main information | body mass index (BMI), arm volume measurement (secondary lymphedema), assessment of tumor recurrence, scale results of Beck Depression Inventory-II |
| Financial information | treatment related cost |

## Sample size calculation

Sample size will be estimated using the sample size estimation formula for prevalence studies. ($N = \frac{\mu_{\frac{z}{2}}^2(1-p)p}{d^2}$) [24, 25]. The formally required sample size calculation was based on the actual incidence of the primary outcomes in the electronic medical record system of West China Hospital. The maximum permissible error is 2%. Based on retrospective data from the electronic medical record system from 2017 to 2020 in West China Hospital, we found a 3-year BCRL incidence rate of 17.57% after breast surgery or breast reconstruction in patients with breast cancer. The rate of BCR occurrence was 15.53% at three years. Assuming a 10% dropout rate, at least 1557 participants should be recruited in our cohort study. We are trying to recruit more breast cancer patients, considering the longevity of the study.

## Statistical analyses

The following primary factors will be calculated when each of the study stages (perioperative stage, chemotherapy therapy stage, radiation therapy stage, follow-up stage) is completed: BCRL incidence, RBC rate, and incidence of depression. To calculate the incidence of the primary outcomes, we will sum the total of participants within each of the stages as the denominator and sum the total of people with one of the primary outcomes as the numerator. For every statistical analysis, the participants will be classified into two groups based on whether they developed secondary lymphedema. Incidence rates of breast cancer recurrence and depression will be calculated separately for groups. The chi-square test will be used to compare the rates rate between the two groups [26]. As a secondary outcome, depression symptoms will be repeatedly measured in this study. The results of the assessment of depression symptoms

will be compared across periods and between treatment-related events using statistical analysis with repeated measures analysis of variance (ANOVA).

We will identify clinicopathological factors, surgery information, lymph node management, breast cancer-related treatment, and body mass index(BMI) as risk factors due to their reported associations with secondary lymphedema, recurrence of breast cancer, and depression symptoms. Multivariate logistic regression will be used to determine whether secondary lymphedema and other parameters can predict breast cancer recurrence. To assess the correlates with these risk factors and outcomes, we will describe the strength of the association using clinically meaningful and statistically significant effect sizes, and a correction will be based on demographic characteristics.

## Missing data

We will assess missing data and record the study stage and reasons for missing data. Data in the previous stage will be retained and calculated if patients are lost to follow-up at this stage of the study. Missing data will not be handled using estimation.

## Availability of data and materials

The data should remain confidential. Only qualified researchers and data analysts can access raw data in our study. All demographic data will be deidentified by allocating numbers to all patients. The datasets that will be used during the study are available on reasonable request by contacting the corresponding author. Dissemination activities will include peer-reviewed publications and international conferences.

## Patient and public involvement

No patient or public is involved.

## Discussion

The outcomes of our cohort study will include the following: (1) Exploratory analysis of breast cancer-related lymphedema to establish the screening program for secondary lymphedema. (2) Exploratory analysis of recurrence of breast cancer to estimate the incidence of recurrence of breast cancer. (3) Continuous assessment of depression during the perioperative, treatment, and follow-up periods. (4) Whole-course analysis of the incidence rate of secondary lymphedema, recurrence of breast cancer, depression, and calculation of hospital and treatment costs. (5) Whether breast cancer survivors with secondary lymphedema have a higher risk of breast cancer recurrence remains unknown. Our study is explorative. We would like to know whether early screening of secondary lymphedema could be an addition to improving early detection and early treatment of breast cancer recurrence. (6) The development of rich related data on breast cancer, including personal characteristics, clinicopathological factors, surgical factors, treatment factors, and financial and mental burdens, can lead to the assessment of the impacts of breast cancer on breast cancer survivors. On review of the literature, there is no previous longitudinal cohort that gives serious consideration to all six points related to breast cancer survivors. However, these factors have direct and indirect effects on health. Therefore, all the participants in our study would benefit from a timely screening program. The results of this study will inform the establishment of a package plan for the early screening of breast cancer survivors.

Among patients with breast cancer treated, we found that being diagnosed with breast cancer-related lymphedema is associated with a modestly increased risk of breast cancer

recurrence [27]. Breast cancer-related lymphedema dramatically impacts the quality of life. Secondary lymphedema, a primary complication, causes the most disability in breast cancer survivors [28]. Cancer recurrence threatens patients' lives. Cancer recurrence drives the most deaths from breast cancer. According to evidence-based data, the presence of new-onset lymphedema should be confirmed to determine whether breast cancer first recurs. Moreover, conservative management is becoming the treatment of choice for secondary lymphedema. However, complete decongestive therapy (the first choice of conservative management) will not be used if the neoplasm is the cause of breast cancer-related lymphedema. Early screening and identification of secondary lymphedema and recurrence of breast cancer are of enormous significance to breast cancer medical prognosis. However, there is no continuous assessment longitudinal cohort established in China. We will focus on the correlation assessment between secondary lymphedema and breast cancer recurrence. This work will be the cornerstone of a well-directed screening program and help to develop a well-defined intervention. In the era of heightened whole-course management, serious complication screening and recurrence identification of breast cancer survivors may be even more important.

We will also focus on the economic and mental burdens of breast cancer [29, 30]. It is associated with poor physical function, role function, emotional function, cognitive function, and social function. This is because depression magnifies the possibility of negative outcomes and spends more cost and time on breast cancer-related treatment. There is no mandated screening program for depression in breast cancer survivors in China. It is unknown whether breast cancer-related treatment and drug costs correlate with an increased risk for depression. As a result, longer follow-up studies of depression and medicine-related costs can provide more information about the long-term economic and mental burden of breast cancer survivors. Moreover, financial stress is well-established as an important reason for depression. The hospital and treatment expenses of breast cancer patients are the slowly draining income of patients and their families. Fear of breast cancer recurrence and financial stress may produce the negative effect of making the person more depressive.

## Limitations

Data from breast cancer patients will only be collected from a large district general hospital. It is difficult to put the scheme of further multicentric trials into practice under the normalization of epidemic prevention and control. Moreover, similar to other cohort studies, the situation in which breast cancer survivors are lost to follow-up seems inevitable. The completeness of treatment and follow-up data will be important to assess the correlation between breast cancer-related lymphedema and recurrence and disease burden (depression and medicine cost). Loss of follow-up will continue to hinder the data completeness of our cohort study.

## Supplements

All the ethical and funding approval documentation has been uploaded as Study Protocol Proofs. We received governmental funding, and the study protocol has undergone peer review by the funding bodies.

Our study is a proposed study that has not completed participant recruitment at the time of submission. No publications containing the results of this study have already been published or submitted to any journal.

## Acknowledgments

The authors thank for the support of the fund, thanks for the help of everybody in our study groups.

## Author Contributions

**Conceptualization:** Huaying Chen, Shaoyong Wu.

**Funding acquisition:** Shaoyong Wu.

**Methodology:** Linli Zhuang, Qian Chen, Xia Liu, Li Liu.

**Project administration:** Huaying Chen, Xia Liu, Shaoyong Wu.

**Supervision:** Xia Liu.

**Validation:** Qian Chen.

**Visualization:** Zhenzhen Feng.

**Writing – original draft:** Linli Zhuang.

**Writing – review & editing:** Linli Zhuang, Xuemei Zheng, Shaoyong Wu, Xiaolin Shen.

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
