## [Decision Letter · Decision Letter 0]

2 Feb 2023

PONE-D-22-27177

Breast cancer-related lymphedema and recurrence of breast cancer: Protocol for a prospective cohort study in China

PLOS ONE

Dear Dr. Wu,

Thank you for submitting your manuscript to PLOS ONE. After careful consideration, we feel that it has merit but does not fully meet PLOS ONE’s publication criteria as it currently stands. Therefore, we invite you to submit a revised version of the manuscript that addresses the points raised during the review process.

We look forward to receiving your revised manuscript.

Kind regards,

Ibrahim Umar Garzali, MBBS, FWACS

Academic Editor

PLOS ONE

Journal Requirements:

2. Our staff editors have determined that your manuscript is likely within the scope of our Early Detection, Screening and Diagnosis of Cancer Call for Papers. This editorial initiative is headed by in-house PLOS editors. This Call for Papers aims to explore recent advances in the early detection of cancer and implications of these advances for patient survival. Additional information can be found on our announcement page: https://collections.plos.org/call-for-papers/early-detection-screening-and-diagnosis-of-cancer/

If you would like your manuscript to be considered for this collection, please let us know in your cover letter and we will ensure that your paper is treated as if you were responding to this call. Please note that being considered for the Call for Papers does not require additional peer review beyond the journal’s standard process and will not delay the publication of your manuscript if it is accepted by PLOS ONE. If you would prefer to remove your manuscript from collection consideration, please specify this in the cover letter.

3. Please amend the manuscript submission data (via Edit Submission) to include author “Linli Zhuang”

“NO authors have competing interests”

7. Your ethics statement should only appear in the Methods section of your manuscript. If your ethics statement is written in any section besides the Methods, please delete it from any other section.

Reviewers' comments:

Reviewer's Responses to Questions

**Comments to the Author**

1. Does the manuscript provide a valid rationale for the proposed study, with clearly identified and justified research questions?

Reviewer #1: Yes

Reviewer #2: No

Reviewer #3: Yes

2. Is the protocol technically sound and planned in a manner that will lead to a meaningful outcome and allow testing the stated hypotheses?

Reviewer #1: Partly

Reviewer #2: No

Reviewer #3: Yes

3. Is the methodology feasible and described in sufficient detail to allow the work to be replicable?

Reviewer #1: Yes

Reviewer #2: No

Reviewer #3: Yes

4. Have the authors described where all data underlying the findings will be made available when the study is complete?

Reviewer #1: Yes

Reviewer #2: No

Reviewer #3: Yes

5. Is the manuscript presented in an intelligible fashion and written in standard English?

Reviewer #1: No

Reviewer #2: No

Reviewer #3: Yes

6. Review Comments to the Author

You may also provide optional suggestions and comments to authors that they might find helpful in planning their study.

Reviewer #1: 2,3. Radical mastectomy is no longer a standard treatment in breast cancer patients.

-Beck Depression Inventory II questionnaire should be placed as an appendix.

-How to measure upper limb volume not clearly mentioned.

-Calculation of sample size should come before the data analysis.

-Some statements should be in future tenses not past tenses.

5.- Introduction not adequately presented the topic and key terms not well defined.

There is need for more literature review.

-Some statements were not well referenced.

-Some part of the introduction should be in discussion not introduction

- There are some grammatical errors.( I have highlighted some of the errors in the PDF manuscript).

-The project context should be in introduction not in the method.

-The referencing should be harmonized to only Vancouver referencing system.

NB: I have highlighted some of the other corrections need to be made in the PDF manuscript I uploaded below.

Reviewer #2: The subject is interesting. However, I've found some potential biases in your protocol. i.e. you've excluded patient with age>55 years. Realistically, older people will have a worse outcome. Furthermore, the risk of depression is higher in elderly. It seems risky to put together several characteristics whose relationship is quite obvious, and amply demonstrated in the Literature.

You highlight the "deep analysis", but it's not clear the way.

Concerning the style, I've noticed that sometimes you use "will" to describe what is foreseen in your protocol, other times "were". It would be better to keep a uniform format.

Reviewer #3: The manuscript needs some grammatical corrections.

7. PLOS authors have the option to publish the peer review history of their article (what does this mean?). If published, this will include your full peer review and any attached files.

Reviewer #1: **Yes: **Saminu Muhammad

Reviewer #2: No

Reviewer #3: No

---

## [Author Response · Author response to Decision Letter 0]

29 Mar 2023

Dear Dr. Ibrahim

Subject: Submission of revised paper PONE-D-22-27177

Thank you for your email dated 2 Feb 2023 enclosing the reviewers’ comments. We have carefully reviewed the comments and have revised the manuscript accordingly. Our responses are given in a point-by-point manner below. Changes to the manuscript are shown in a separate file labeled “Revised manuscript with Track Changes”.

 We hope the revised version is now suitable for publication and look forward to hearing from you in due course.

Sincerely,

Linli Zhuang

West China Hospital

Response to Reviewer: Thank you for your review of our paper. We have answered each of your points below.

Response: Thanks very much for the suggestions from reviewer. Manuscript and file name had been changed according to journal format. We hope the revised version fully meets PLOS ONE's style requirements . Thank you very much. 

2.Our staff editors have determined that your manuscript is likely within the scope of our Early Detection, Screening and Diagnosis of Cancer Call for Papers. This editorial initiative is headed by in-house PLOS editors. This Call for Papers aims to explore recent advances in the early detection of cancer and implications of these advances for patient survival. If you would like your manuscript to be considered for this collection, please let us know in your cover letter and we will ensure that your paper is treated as if you were responding to this call. Please note that being considered for the Call for Papers does not require additional peer review beyond the journal’s standard process and will not delay the publication of your manuscript if it is accepted by PLOS ONE. If you would prefer to remove your manuscript from collection consideration, please specify this in the cover letter.

Response: Thank you very much. I am glad to hear our manuscript could be considered for this collection. We added the related information to the cover letter. We believe that early detection, screening and diagnosis of cancer will improve health and quality of life of cancer patients. So, we are glad to responding to this call. Thank you. 

3. Please amend the manuscript submission data (via Edit Submission) to include author “Linli Zhuang”

Response: Thank you very much. I have amended the manuscript submission data(via Edit Submission) to include author “Linli Zhuang”. Thank you.

4. Thank you for stating the following in your Competing Interests section:“NO authors have competing interests”. Please complete your Competing Interests on the online submission form to state any Competing Interests. If you have no competing interests, please state "The authors have declared that no competing interests exist.", as detailed online in our guide for authors. This information should be included in your cover letter; we will change the online submission form on your behalf.

Response: Thank you very much. We completed our competing interests on the online submission form accordingly. And we also added the competing interests section in our cover letter. Thank you.

.

5. In your Data Availability statement, you have not specified where the minimal data set underlying the results described in your manuscript can be found. PLOS defines a study's minimal data set as the underlying data used to reach the conclusions drawn in the manuscript and any additional data required to replicate the reported study findings in their entirety. All PLOS journals require that the minimal data set be made fully available. For more information about our data policy, please see http://journals.plos.org/plosone/s/data-availability. Upon re-submitting your revised manuscript, please upload your study’s minimal underlying data set as either Supporting Information files or to a stable, public repository and include the relevant URLs, DOIs, or accession numbers within your revised cover letter. For a list of acceptable repositories, please see http://journals.plos.org/plosone/s/data-availability#loc-recommended-repositories. Any potentially identifying patient information must be fully anonymized.Important: If there are ethical or legal restrictions to sharing your data publicly, please explain these restrictions in detail. Please see our guidelines for more information on what we consider unacceptable restrictions to publicly sharing data: http://journals.plos.org/plosone/s/data-availability#loc-unacceptable-data-access-restrictions. Note that it is not acceptable for the authors to be the sole named individuals responsible for ensuring data access. We will update your Data Availability statement to reflect the information you provide in your cover letter. 

Response: Thank you very much. However, our paper is only a protocol. We have not begun to collect the data yet. Our study is not involved minimal data set at this time. Thanks.

Response: Thank you very much. I have revised the contents of this part. We already included captions for our supporting information files at the end of manuscript, and update any in-text citations to match accordingly. Thanks.

7. Your ethics statement should only appear in the Methods section of your manuscript. If your ethics statement is written in any section besides the Methods, please delete it from any other section. 

Response: Thank you very much. Your suggestion pointed out exactly where we had gone wrong I have deleted ethics statement from any other section in our article. Thank you very much.

Response: Thank you very much. I have added the additional references and checked the reference list to ensure that it is complete and correct (Line 340-431). I have revised the content of this part. Thank you. 

9. Review Comments to the Author

Please use the space provided to explain your answers to the questions above and, if applicable, provide comments about issues authors must address before this protocol can be accepted for publication. You may also include additional comments for the author, including concerns about research or publication ethics. You may also provide optional suggestions and comments to authors that they might find helpful in planning their study.

(1) Reviewer #1: 2,3. Radical mastectomy is no longer a standard treatment in breast cancer patients.

Response: Thank you for your question. Our research is an exploratory clinical research. Different types of breast cancer surgery have varying influences in breast cancer related lymphedema(one of main outcomes). Therefore, other types of breast cancer surgery do not meet the purpose of our research. The participants of research in this study were patients who will suffer radical mastectomy only. Further research is needed to extend the findings to other types of breast cancer surgery. Thank you very much.

(2) Beck Depression Inventory II questionnaire should be placed as an appendix.

Response: Thank you for your suggestions. The Chinese version of Beck Depression Inventory II questionnaire was used in this study. I have added Beck Depression Inventory II questionnaire (Chinese version) as an appendix of our study(Line439-440). Thank you very much.

(3)How to measure upper limb volume not clearly mentioned.

Response: Thank you for your question. I am very sorry that this part was not clear in the original manuscripts. The upper-limb volume measurement is based on the international standard measurement method according to 2020 Consensus Document of the International Society of Lymphology. We a device commercially known as the Perometer. Its basic operating principle is illustrated in Figure 1 below.

Figure 1 

Figure 1 Illustrating automated optoelectronic measurement of arm circumferences. A limb (arm or leg) is placed within a movable frame that contains infrared light sources that illuminate the limb and allow acquisition of limb projected perpendicular dimensions 

Measurement Details and Volume Calculations

A sliding frame with imbedded infrared (IR) light sources scans the arm and the “shadow” dimensions D1 and D2 are detected and used calculate cross-sectional areas as a constant (k) multiplied by D1 and D2. Segment volumes are determined similarly to the manual method with segment volumes summed to produce the arm volume of interest. This method has the advantage of rapidly estimating cross-sectional areas using as low as 0.5 cm segment lengths and an automatic calculation of arm volumes. 

We expounded the measurement in the revised manuscripts(Line 150-152). Thank you very much.

(4)Calculation of sample size should come before the data analysis.

Response: Thank you for your suggestions. We adapted the order in the methods accordingly(Line 216-225). Thank you very much.

(5)Some statements should be in future tenses not past tenses.

Response: Thank you for your suggestions. We have changed the past tense to future tense throughout the document, as appropriate. Thank you very much.

(6)Introduction not adequately presented the topic and key terms not well defined.

Response: Thank you for your suggestions. The introduction section has now been enriched(Line 65-93). And we give a more precise definition of key terms. we can learn a lot from you. Thank you.

(7)There is need for more literature review.

Response: Thank you for your suggestions. We added more literature review to enrich our paper. We believe that more literature review could make our article clear and readable. Thank you very much.

(8)Some statements were not well referenced.

Response: Thank you for your suggestions. We added references to support the statements. Thank you very much.

(9)Some part of the introduction should be in discussion not introduction

Response: Thank you for your suggestions. We have moved some contents of introduction to the discussion section(Line 84-93). Thank you very much.

(10)There are some grammatical errors.( I have highlighted some of the errors in the PDF manuscript).

Response: Thank you for your suggestions. You have highlighted the error points in yellow in the PDF manuscript which pointed out exactly where we had gone wrong. I have revised the contents accordingly. we upload revised PDF manuscript as a separate file labeled 'Lighlighted for Corrections(reply version)'. Thank you very much. 

(11)The project context should be in introduction not in the method.

Response: Thank you for your suggestions. We have moved project contents of method to the introduction section(Line 84-88). Thank you very much.

(12)-The referencing should be harmonized to only Vancouver referencing system.

NB: I have highlighted some of the other corrections need to be made in the PDF manuscript I uploaded below.

Response: Thank you for your suggestions. I have checked the references list to ensure that it is complete and correct. I also have revised all the corrections which you pointed in the PDF manuscript. We reply in full in the revised PDF manuscript. The PDF manuscript with replies was uploaded as supplementary documents. Thank you very much.

(13)Reviewer #2: The subject is interesting. However, I've found some potential biases in your protocol. i.e. you've excluded patient with age>55 years. Realistically, older people will have a worse outcome. Furthermore, the risk of depression is higher in elderly. It seems risky to put together several characteristics whose relationship is quite obvious, and amply demonstrated in the Literature.

Response: Thank you for your question. In clinical studies, older adults are one of vulnerable groups. Vulnerable groups (e.g., children, elderly, people with disability, residents with chronic disease, economically disadvantaged and/or isolated people) should be particularly protected. Our study is an exploratory study. We hope we can draw some conclusions from adults group first. 

And then these conclusion will be generalized to other groups. Therefore, our clinical trials protocol did not include vulnerable populations, as patient with age>55 years. And the further study will extends these research results in the population of older adults. Thank you very much.

(14)You highlight the "deep analysis", but it's not clear the way.

Response: Thank you for your question. I am sorry for inaccurate description. The part of this article is an exploratory analysis. Confirmatory analysis is not carried out(Line 260, 262). Thank you very much.

(15)Concerning the style, I've noticed that sometimes you use "will" to describe what is foreseen in your protocol, other times "were". It would be better to keep a uniform format.

Response: Thank you for your suggestions. We have changed the past tense to future tense throughout the document, as appropriate. Thank you very much.

(16)Reviewer #3: The manuscript needs some grammatical corrections.

Response: Thank you for your suggestions. We have corrected the grammatical errors. Thank you very much.

We tried our best to improve the manuscript and made some changes in the manuscript. These changes will not influence the content and framework of the paper. And here we did not list the changes but marked in yellow in revised paper.

We appreciate for Editors/Reviewers’ warm work earnestly, and hope that the correction will meet with approval.

Once again, thank you very much for your comments and suggestions.

---

## [Editor Report · Decision Letter 1]

2 May 2023

Breast cancer-related lymphedema and recurrence of breast cancer: Protocol for a prospective cohort study in China

PONE-D-22-27177R1

Dear Dr. Wu,

We’re pleased to inform you that your manuscript has been judged scientifically suitable for publication and will be formally accepted for publication once it meets all outstanding technical requirements.

Kind regards,

Ibrahim Umar Garzali, MBBS, FWACS

Academic Editor

PLOS ONE
---

## [Editor Report · Acceptance letter]

5 May 2023

PONE-D-22-27177R1 

Breast cancer-related lymphedema and recurrence of breast cancer: Protocol for a prospective cohort study in China 

Dear Dr. Zhuang:

I'm pleased to inform you that your manuscript has been deemed suitable for publication in PLOS ONE. Congratulations! Your manuscript is now with our production department. 

Kind regards, 

on behalf of

Dr. Ibrahim Umar Garzali 

Academic Editor

PLOS ONE